# Adipose tissue biomarkers and type 2 diabetes incidence in normoglycemic participants in the MESArthritis Ancillary Study: A cohort study

Farhad Pishgar[1], Mahsima Shabani[2], Thiago Quinaglia A. C. Silva[2], David A. Bluemke[3], Matthew Budoff[4], R Graham Barr[5,6], Matthew A. Allison[7], Alain G. Bertoni[8], Wendy S. Post[2], João A. C. Lima[2], Shadpour Demehri[1] *

1 Russell H. Morgan Department of Radiology and Radiological Science, Johns Hopkins University School of Medicine, Baltimore, Maryland, United States of America, 2 Division of Cardiology, Department of Medicine, Johns Hopkins University School of Medicine, Baltimore, Maryland, United States of America, 3 Department of Radiology, University of Wisconsin School of Medicine and Public Health, Madison, Wisconsin, United States of America, 4 Lundquist Institute at Harbor-University of California Los Angeles School of Medicine, Torrance, California, United States of America, 5 Department of Medicine, Columbia University Medical Center, New York, New York, United States of America, 6 Department of Epidemiology, Columbia University Medical Center, New York, New York, United States of America, 7 Division of Preventive Medicine, Department of Family Medicine and Public Health, University of California San Diego, La Jolla, California, United States of America, 8 Division of Public Health Sciences, Wake Forest School of Medicine, Winston-Salem, North Carolina, United States of America

* Demehri2001@Yahoo.com, SDemehr1@JHMI.edu

**Data Availability Statement:** Data are available at MESA-NHLBI.org, upon receiving approval from the Multi-Ethnic Study of Atherosclerosis (MESA)

## Abstract

### Background

Given the central role of skeletal muscles in glucose homeostasis, deposition of adipose depots beneath the fascia of muscles (versus subcutaneous adipose tissue [SAT]) may precede insulin resistance and type 2 diabetes (T2D) incidence. This study was aimed to investigate the associations between computed tomography (CT)–derived biomarkers for adipose tissue and T2D incidence in normoglycemic adults.

### Methods and findings

This study was a population-based multiethnic retrospective cohort of 1,744 participants in the Multi-Ethnic Study of Atherosclerosis (MESA) with normoglycemia (baseline fasting plasma glucose [FPG] less than 100 mg/dL) from 6 United States of America communities. Participants were followed from April 2010 and January 2012 to December 2017, for a median of 7 years. The intermuscular adipose tissue (IMAT) and SAT areas were measured in baseline chest CT exams and were corrected by height squared (SAT and IMAT indices) using a predefined measurement protocol. T2D incidence, as the main outcome, was based on follow-up FPG, review of hospital records, or self-reported physician diagnoses.

Participants' mean age was 69 ± 9 years at baseline, and 977 (56.0%) were women. Over a median of 7 years, 103 (5.9%) participants were diagnosed with T2D, and 147 (8.4%) participants died. The IMAT index (hazard ratio [HR]: 1.27 [95% confidence interval

steering committee and local institutional ethics board(s). Please see MESA-NHLBI.org/Publications.aspx or email MESA Coordinating Center at chsccweb@u.washington.edu for more details.

**Funding:** The MESA study was supported by contracts 75N92020D00001, HHSN268201500003I, N01-HC-95159, 75N92020D00005, N01-HC-95160, 75N92020D00002, N01-HC-95161, 75N92020D00003, N01-HC-95162, 75N92020D00006, N01-HC-95163, 75N92020D00004, N01-HC-95164, 75N92020D00007, N01-HC-95165, N01-HC-95166, N01-HC-95167, N01-HC-95168, N01-HC-95169, R01-HL-077612, and R01-HL-093081 from the National Heart, Lung, and Blood Institute (NHLBI), and by grants UL1-TR-000040, UL1-TR-001079, and UL1-TR-001420 from the National Center for Advancing Translational Sciences (NCATS). This publication was developed under a STAR research assistance agreement, No. RD831697 (MESA Air) and RD-83830001 (MESA Air Next Stage), awarded by the U.S Environmental Protection Agency (EPA). It has not been formally reviewed by the EPA. The views expressed in this document are solely those of the authors and the EPA does not endorse any products or commercial services mentioned in this publication. The aforementioned funding agencies provided financial support of the study for participant recruitment and data collection (paid to the MESA field centers). The funders had no role in study design, data collection and analysis, decision to publish, or preparation of the manuscript. The specific roles of the authors affiliated with the MESA field centers are articulated in the 'author contributions' section.

**Competing interests:** The authors have read the journal's policy and have the following competing interest(s): Relevant to this study, RGB received grant funding from the National Institutes of Health (NIH) and the Chronic Obstructive Pulmonary Disease (COPD) Foundation. Other authors have declared that no competing interests exist.

**Abbreviations:** ASCVD, atherosclerotic cardiovascular disease; BMI, body mass index; CI, confidence interval; CT, computed tomography; FPG, fasting plasma glucose; HDL, high-density lipoprotein; HOMA-IR, homeostatic model assessment–insulin resistance; HR, hazard ratio; HU, Hounsfield unit; IMAT, intermuscular adipose tissue; MESA, Multi-Ethnic Study of Atherosclerosis; MET, metabolic equivalent; PM, pectoralis muscle; SAT, subcutaneous adipose tissue; SD, standard deviation; STROBE,

[CI]: 1.15–1.41] per 1-standard deviation [SD] increment) and the SAT index (HR: 1.43 [95% CI: 1.16–1.77] per 1-SD increment) at baseline were associated with T2D incidence over the follow-up. The associations of the IMAT and SAT indices with T2D incidence were attenuated after adjustment for body mass index (BMI) and waist circumference, with HRs of 1.23 (95% CI: 1.09–1.38) and 1.29 (95% CI: 0.96–1.74) per 1-SD increment, respectively. The limitations of this study include unmeasured residual confounders and one-time measurement of adipose tissue biomarkers.

## Conclusions

In this study, we observed an association between IMAT at baseline and T2D incidence over the follow-up. This study suggests the potential role of intermuscular adipose depots in the pathophysiology of T2D.

## Trial registration

ClinicalTrials.gov NCT00005487

## Author summary

### Why was this study done?

- This study was designed to investigate the associations between computed tomography (CT)–derived biomarkers for adipose tissue at baseline and type 2 diabetes (T2D) incidence in normoglycemic adults.

### What did the researchers do and find?

- We assessed 1,744 normoglycemic participants with chest CT exams between 2010 and 2012. The intermuscular adipose tissue (IMAT) and subcutaneous adipose tissue (SAT) areas were measured in these chest CT exams using a predefined measurement protocol.

- Participants were followed from April 2010 and January 2012 to December 2017, for a median of 7 years, for T2D incidence. T2D incidence was based on follow-up fasting plasma glucose (FPG), review of hospital records, or self-reported physician diagnoses.

- This study found that a higher CT-derived IMAT at baseline was associated with T2D incidence over the follow-up.

### What do these findings mean?

- In normoglycemic participants, the IMAT deposition was associated with T2D incidence. This study suggests the potential role of intermuscular adipose depots in the pathophysiology of T2D.

Strengthening the Reporting of Observational Studies in Epidemiology; T2D, type 2 diabetes.

- The CT-derived adipose tissue biomarkers are obtainable from CT exams performed for other initial indications and can extend the value of the routinely performed chest CT exams. Such biomarkers may be associated with T2D incidence.

## Introduction

Type 2 diabetes (T2D) affects more than 20 million new cases every year, and, in addition to the attributable high morbidity and mortality, imposes an increased financial burden on healthcare systems worldwide [1]. However, most of this burden can be avoided by early identification of at-risk individuals and the rapid implementation of primary and secondary preventive measures [2]. Therefore, identifying the biomarkers associated with T2D incidence is crucial to minimize the morbidity, mortality, and healthcare financial burden attributable to this prevalent and chronic disease.

There are known associations between obesity or excessive overall adipose depots and the T2D incidence [3–5]. However, recent data from genome-wide association studies [6,7] and imaging assessments [7] suggest a stronger association between ectopic adipose depots (in liver, visceral organs, and muscles) and T2D incidence (compared to excessive overall adipose depots) [8–10]. Specifically, adipose depots deposition beneath the fascia of skeletal muscles (extramyocellular and intramyocellular lipid content) may contribute to T2D incidence, owing to the skeletal muscles cardinal role in glucose homeostasis [11]. The body mass index (BMI) and other clinical anthropometric indices, despite quantifying the excessive overall adipose depots, do not provide data on ectopic adipose depots distribution.

Imaging-derived adipose tissue biomarkers have the potential to provide a better characterization of ectopic adipose depots distribution [12]. In addition to dedicated imaging techniques for evaluating body composition and adipose depots distribution (e.g., dual-energy X-ray absorptiometry or bioelectrical impedance), chest computed tomography (CT) exams that are commonly performed in the routine clinical practice for cardiopulmonary indications (e.g., coronary calcium scoring [13] or lung cancer screening [14]) retain data on adipose depots distribution, and there is an opportunity to extract biomarkers on the distribution of ectopic adipose depots from these CT exams, at zero additional cost or radiation exposure [15–17].

To our knowledge, no prior studies assessed the associations between CT-derived adipose depots biomarkers and T2D incidence using longitudinal analysis [3,8–10] after adjusting for the effects of traditional risk factors of the disease and clinical anthropometric indices (e.g., BMI and waist circumference) [8,9] or stratified for baseline glycemic status (normoglycemia versus prediabetes) [9].

In the present study, we used the population-based multiethnic cohort of participants in the Multi-Ethnic Study of Atherosclerosis (MESA) to assess the associations between CT-derived adipose depots biomarkers and T2D incidence. We characterized intermuscular adipose tissue (IMAT), subcutaneous adipose tissue (SAT), and intramyocellular lipid contents in the baseline chest CT exams and studied the potential associations between these biomarkers and T2D incidence.

## Methods

The MESA is a population-based multiethnic cohort of 6,814 participants from 6 communities across the USA to investigate the features of subclinical and clinical cardiovascular diseases

and to determine their relevant risk factors (see www.MESA-NHLBI.org) [18]. The MESA was approved by the institutional review boards of the 6 participating field centers (Columbia University, Johns Hopkins University, Northwestern University, University of California, University of Minnesota, and Wake Forest University) and the coordinating center (University of Washington). All participants in the MESA provided written informed consent (registered at ClinicalTrials.gov as NCT00005487).

## MESArthritis Ancillary Study

Between April 2010 and January 2012 (fifth MESA exam), 3,137 participants in the MESA consented to participate in the MESA Lung Ancillary Study and underwent chest CT exams [19]. The MESArthritis Ancillary Study is an analysis of the available CT exams of 3,083 of these participants [20]. It is aimed to investigate the roles of CT-derived soft tissue and bone biomarkers associated with incidence and clinical outcomes of several cardiometabolic and cardiopulmonary diseases. Before this study, 86 participants with low-quality chest CT exams ($n$ = 52), unknown baseline glycemic status ($n$ = 21), or missing follow-up information ($n$ = 13) were excluded (S1 Fig).

In the baseline exam, information on various demographic and clinical characteristics was collected. Participants were visited to obtain plasma samples after 12 hours of fasting to measure plasma glucose, insulin, HbA1c, triglyceride, and high-density lipoprotein (HDL) cholesterol [21,22]. In this study, normoglycemic participants (baseline fasting plasma glucose [FPG] less than 100 mg/dL) were included (S1 Fig; participants with prediabetes, FPG of 100 to 125 mg/dL, were included in a supplementary analysis).

## T2D diagnosis

Between baseline (April 2010 to January 2012) to December 2017, participants were contacted by interviewers at intervals of 9 to 12 months to inquire about new disease diagnoses and interim hospital admissions. Moreover, between September 2016 and June 2018, participants completed a follow-up exam to recollect information on clinical characteristics and obtain fasting plasma samples (sixth MESA exam). This information was supplemented by the data collected through reviews of the hospital records (S2 Fig).

T2D diagnosis over the follow-up was based on at least 1 of the following criteria: (1) physician-diagnosed T2D based on the review of the hospital records (based on the relevant codes of the ninth and 10th editions of the International Classification of Diseases, S1 Table, time of incidence was the time of hospital admission); (2) self-reported physician-diagnosed T2D (time of incidence was the midpoint between the last interview without and the interview with the self-reported physician-diagnosed disease); or (3) use of insulin or oral hypoglycemic agents or FPG ≥126 mg/dL in the follow-up exam (time of incidence was the time of the follow-up exam).

In a sensitivity analysis, self-reported physician-diagnosed T2D (the second criterion) was confirmed with the use of insulin or oral hypoglycemic agents or FPG ≥126 mg/dL in the follow-up exam, and participants with self-reported T2D but missing information on the use of insulin or oral hypoglycemic agents or FPG in the follow-up exam ($n$ = 17) were excluded.

## Adipose tissue biomarkers

The non-contrast–enhanced chest CT exams (acquired at suspended full inspiration using 64-slice multidetector row CT scanners, Siemens Medical Solutions, Erlangen, Germany or GE Healthcare, Waukesha, Wisconsin, USA) were used to measure the IMAT and SAT areas as well as the pectoralis muscles (PMs) density [19,23]. The cross-sectional SAT area was

measured as the area between the PM and skin surface in the slice just above the superior margin of the aortic arch [20,24]. Density of the SAT area was analyzed to estimate an individualized attenuation threshold for the IMAT (i.e., the extramyocellular lipid content) [20,25]. The cross-sectional areas within the PM with attenuation below the estimated threshold were measured as the IMAT (S3 Fig). The IMAT and SAT areas were corrected as IMAT and SAT indices (area by height squared, $cm^2/m^2$) to account for the anthropometric variations. Moreover, the mean density of the PM area (after excluding the IMAT area) was also measured as a surrogate measure of the intramyocellular lipid content, Hounsfield unit by area, $HU/cm^2$, S3 Fig).

## Statistical analysis

This study is reported as per the Strengthening the Reporting of Observational Studies in Epidemiology (STROBE) guidelines (S1 Checklist). The statistical analyses of this study were planned a priori at the time of preparing the research proposal and was approved by the MESA publication and steering committees. Few additions (e.g., assessing the linearity of the associations between adipose tissue biomarkers and T2D incidence and multiply imputing the missing data points) or changes (e.g., reporting the associations per 1-standard deviation [SD] increment in the biomarkers) were made to the planned statistical analyses at the time of data analysis or per peer reviewers comments (S2 Table).

Descriptive statistics were used to compare baseline demographic and clinical characteristics between normoglycemic participants with and without T2D incidence over the follow-up.

The correlations between adipose tissue biomarkers with baseline BMI, waist circumference, triglyceride, HDL cholesterol, FPG, insulin, HbA1c, and homeostatic model assessment–insulin resistance (HOMA-IR, FPG × insulin/405) were analyzed using the Pearson correlation analysis and illustrated using a symmetric correlation matrix.

The generalized additive Cox proportional hazard models with integrated smoothness estimation were used to assess and illustrate the linearity of the associations between adipose tissue biomarkers and T2D incidence. Moreover, several adjusted Cox proportional hazard models were used to estimate the hazard ratios (HRs) and 95% confidence interval (CI) for the T2D incidence according to adipose tissue biomarkers (per 1-SD increment). Models were tested for the proportional hazard assumption by regressing the Schoenfeld residuals over time. Models were adjusted for the traditional risk factors of T2D, HOMA-IR, and clinical anthropometric indices (i.e., BMI and waist circumference) at baseline. The traditional risk factors of T2D included age, sex, race/ethnicity, smoking status, alcohol drinking status, physical activity (vigorous and moderate metabolic equivalents [METs]), HDL cholesterol, triglyceride, and hypertension [3]. Decedents without T2D incidence were right-censored at the time of death.

Stratified analyses for the traditional risk factors of T2D, HOMA-IR, BMI, and waist circumference were conducted. In the HOMA-IR stratified analyses, the mean HOMA-IR in normoglycemic participants without T2D incidence over the follow-up was used as the cut point. High HDL cholesterol, high triglyceride, hypertension, and central obesity were defined according to the updated metabolic syndrome guideline of the National Cholesterol Education Program Adult Treatment Panel III [26]. Heterogeneity of the association between adipose tissue biomarkers and T2D incidence in the levels of the stratification variable was tested using the significance of a multiplicative interaction term between adipose tissue biomarkers and the stratification variable.

In a sensitivity analysis, self-reported physician-diagnosed T2D (the second criterion) was confirmed with the use of insulin or oral hypoglycemic agents or FPG ≥126 mg/dL in the follow-up exam, and similar Cox proportional hazard models were used to study the associations between adipose tissue biomarkers and the T2D incidence.

In a supplementary analysis, similar Cox proportional hazard models were used to study the associations between adipose tissue biomarkers and the T2D incidence in participants with prediabetes at baseline.

The missing data points were multiply-imputed with chained equations and predictive mean matching method before Cox proportional hazard modeling and the stratified analyses to produce 5 datasets [27] Each dataset was analyzed separately, and the results were pooled across the datasets using Rubin's rule (missing values were infrequent among the traditional risk factors of T2D, **S4 Fig**).

We applied the Benjamini–Hochberg procedure to correct the *p*-values for multiple comparisons. The *p*-values from main, sensitivity, and supplementary analyses were batched and corrected separately using this procedure [28]. The associations with *p*-values <0.05 were considered statistically significant. All analyses were performed in the R platform, version 3.6.1 (R Foundation for Statistical Computing, Vienna, Austria).

## Results

### Baseline characteristics

Out of the 3,083 participants in the MESArthritis Ancillary Study, 1,744 normoglycemic participants at baseline were included in this study (**S1 Fig**). Participants' mean age was 68.7 ± 9.3 years at baseline, and 977 (56.0%) participants were female (**Table 1**). Participants were followed for a median of 6.8 [6.2 to 7.2] years, and during this period, 103 (5.9%) participants were diagnosed with T2D, and 147 (8.4%) participants died (137 [7.9%] without T2D, **Fig 1**).

### Intermuscular adiposity and T2D incidence

IMAT index correlated with BMI (r: 0.41, *p*-value: <0.001), waist circumference (r: 0.35, *p*-value: <0.001), HOMA-IR (r: 0.21, *p*-value: <0.001), and the HDL cholesterol (r: −0.15, *p*-value: <0.001) at baseline (**S5 Fig**).

The generalized additive Cox proportional hazard models showed little evidence for a non-linear association between the IMAT index and T2D incidence (**S6 Fig**). In the models adjusted for traditional risk factors of T2D, higher IMAT index quartiles were associated with T2D incidence, and 1-SD increment in the IMAT index was associated with the disease incidence (HR: 1.27 [95% CI: 1.15 to 1.41], *p*-value <0.001, per 1-SD increment). This association was attenuated but remained statistically significant after adjusting for the effects of the HOMA-IR (HR: 1.26 [95% CI: 1.13 to 1.40], *p*-value: <0.001, per 1-SD increment) or BMI and waist circumference (HR: 1.23 [95% CI: 1.09 to 1.38], *p*-value: 0.010, per 1-SD increment) (**Table 2**).

In the stratified analyses, our models showed similar associations between IMAT index and T2D incidence in the strata of the traditional risk factors of T2D, HOMA-IR, BMI, and waist circumference (**Fig 2**, **S7 Fig**).

### Subcutaneous adiposity and T2D incidence

The SAT index correlated with BMI (r: 0.65, *p*-value: <0.001), waist circumference (r: 0.46, *p*-value: <0.001), and HOMA-IR (r: 0.34, *p*-value: <0.001) at baseline (**S5 Fig**).

The generalized additive Cox proportional hazard models showed little evidence for a non-linear association between the SAT index and T2D incidence (**S6 Fig**). Higher SAT index quartiles were associated with T2D incidence, and 1-SD increment in the SAT index was associated with T2D incidence (HR: 1.43 [95% CI: 1.16 to 1.77], *p*-value: 0.010, per 1-SD increment) after adjusting for traditional risk factors of T2D. After including the HOMA-IR (HR:

**Table 1. Baseline demographic and clinical characteristics of normoglycemic participants.**

| Characteristics | Without T2D incidence over the follow-up ($n$ = 1,641) | With T2D incidence over the follow-up ($n$ = 103) |
|---|---|---|
| **Traditional risk factors of T2D** | | |
| Age (years) | 68.7 ± 9.3 | 68.7 ± 9.5 |
| Sex (female) | 924 (56.3%) | 53 (51.5%) |
| Race/ethnicity | | |
| White | 750 (45.7%) | 35 (34.0%) |
| Black | 418 (25.5%) | 30 (29.1%) |
| Hispanic | 279 (17.0%) | 28 (27.2%) |
| Chinese-American | 194 (11.8%) | 10 (9.7%) |
| Physical activity (METs hours/week) | 88.6 ± 96.5 | 85.1 ± 92.1 |
| Alcohol drinking status (current) | 768 (46.8%) | 32 (31.1%) |
| Smoking status | | |
| Never | 757 (46.1%) | 42 (40.8%) |
| Former | 747 (45.5%) | 51 (49.5%) |
| Current | 127 (7.7%) | 10 (9.7%) |
| Systolic blood pressure (mm Hg) | 122.2 ± 19.8 | 125.6 ± 21.3 |
| Diastolic blood pressure (mm Hg) | 68.4 ± 9.9 | 70.0 ± 9.2 |
| TG (mg/dL) | 100.8 ± 52.5 | 118.4 ± 64.7 |
| HDL cholesterol (mg/dL) | 58.5 ± 17.2 | 52.4 ± 15.2 |
| **HOMA-IR** | | |
| $\log_2$ (HOMA-IR) | 3.2 ± 0.9 | 3.6 ± 1.0 |
| **Clinical anthropometric indices** | | |
| BMI (kg/m$^2$) | 27.4 ± 5.1 | 29.7 ± 5.6 |
| Waist C. (cm) | 96.1 ± 13.4 | 100.7 ± 13.0 |
| **Adipose tissue biomarkers** | | |
| IMAT index (cm$^2$/m$^2$) | 0.3 ± 0.3 | 0.5 ± 0.7 |
| SAT index (cm$^2$/m$^2$) | 18.8 ± 11.2 | 22.5 ± 14.6 |
| PM density (HU/cm$^2$) | 24.0 ± 10.3 | 23.4 ± 11.1 |

Quantitative variables are shown in mean ± SD, and qualitative variables are shown in number (%).

BMI, body mass index; HDL, high-density lipoprotein; HOMA-IR, homeostatic model assessment–insulin resistance; HU, Hounsfield unit; IMAT, intermuscular adipose tissue; IQR, interquartile range; MET, metabolic equivalent; PM, pectoralis muscle; SAT, subcutaneous adipose tissue; SD, standard deviation; T2D, type 2 diabetes; TG, triglyceride; Waist C., waist circumference.

1.34 [95% CI: 1.07 to 1.68], $p$-value: 0.078, per 1-SD increment) or BMI and waist circumference (HR: 1.29 [95% CI: 0.96 to 1.74], $p$-value: 0.325, per 1-SD increment), the associations between the SAT index and T2D incidence were attenuated toward null hypothesis and no longer were statistically significant (**Table 2**).

In the stratified analyses, our models showed similar associations between SAT index and T2D incidence in the strata of the traditional risk factors of T2D, HOMA-IR, BMI, and waist circumference (**Fig 2**, **S8 Fig**).

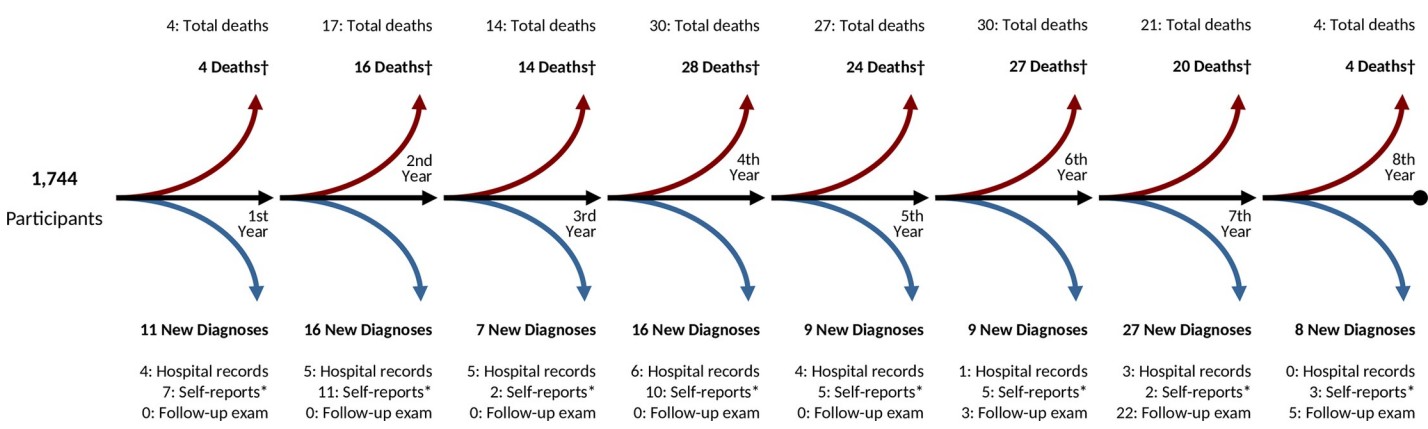

**Fig 1. Flow diagram of participants and the timing of T2D diagnoses.** *Self-reported physician-diagnosed T2D. †Decedents without T2D at the time of death. FPG, fasting plasma glucose; T2D, type 2 diabetes.

## Intramuscular adiposity and T2D incidence

The PM density correlated with BMI (r: −0.33, p-value: <0.001), waist circumference (r: −0.28, p-value: <0.001), and HOMA-IR (r: −0.16, p-value: <0.001) at baseline (S5 Fig).

The generalized additive Cox proportional hazard models showed little evidence for a non-linear association between PM density and T2D incidence (S6 Fig). The models adjusted for

**Table 2. Associations of adipose tissue biomarkers and T2D incidence.**

| | Index | | | | p-value for trend | HR (95% CI), p-value per 1-SD increment |
|---|---|---|---|---|---|---|
| | Quartile 1 | Quartile 2 | Quartile 3 | Quartile 4 | | |
| **IMAT index** | | | | | | |
| Mean (cm²/m²) | 0.1 | 0.2 | 0.3 | 0.7 | - | - |
| Incident cases | 16 | 19 | 26 | 42 | - | - |
| Incidence rate (per 1,000 PYs) | 5.9 | 6.9 | 9.6 | 15.6 | - | - |
| HR (95% CI) | | | | | | |
| Model 0 | 1 (reference) | 1.18 (0.60–2.31) | 1.62 (0.86–3.04) | 2.66 (1.49–4.77) | <0.001 | 1.27 (1.17–1.38), <0.001 |
| Model 1 | 1 (reference) | 1.11 (0.55–2.25) | 1.37 (0.70–2.69) | 1.93 (1.03–3.64) | 0.100 | 1.27 (1.15–1.41), <0.001 |
| Model 1 + HOMA-IR | 1 (reference) | 1.08 (0.54–2.18) | 1.24 (0.63–2.45) | 1.72 (0.91–3.26) | 0.216 | 1.26 (1.13–1.40), <0.001 |
| Model 1 + BMI and Waist C. | 1 (reference) | 0.98 (0.48–1.99) | 1.13 (0.56–2.27) | 1.42 (0.71–2.86) | 0.485 | 1.23 (1.09–1.38), 0.010 |
| **SAT index** | | | | | | |
| Mean (cm²/m²) | 7.6 | 13.1 | 20.4 | 35.1 | - | - |
| Incident cases | 19 | 22 | 26 | 36 | - | - |
| Incidence rate (per 1,000 PYs) | 7.3 | 8.0 | 9.3 | 13.2 | - | - |
| HR (95% CI) | | | | | | |
| Model 0 | 1 (reference) | 1.09 (0.58–2.02) | 1.25 (0.69–2.28) | 1.76 (1.00–3.09) | 0.150 | 1.27 (1.08–1.50), 0.043 |
| Model 1 | 1 (reference) | 1.09 (0.57–2.07) | 1.53 (0.78–3.00) | 2.65 (1.22–5.77) | 0.078 | 1.43 (1.16–1.77), 0.010 |
| Model 1 + HOMA-IR | 1 (reference) | 1.00 (0.53–1.91) | 1.26 (0.62–2.52) | 2.02 (0.90–4.54) | 0.300 | 1.34 (1.07–1.68), 0.078 |
| Model 1 + BMI and Waist C. | 1 (reference) | 0.98 (0.50–1.90) | 1.26 (0.59–2.69) | 1.79 (0.68–4.71) | 0.495 | 1.29 (0.96–1.74), 0.325 |

Model 0: unadjusted.

Model 1: adjusted for categorical age, sex, race/ethnicity, smoking status, alcohol drinking status, physical activity, TG, HDL cholesterol, and hypertension.

Reported p-values were corrected for multiple comparisons.

BMI, body mass index; CI, confidence interval; HDL, high-density lipoprotein; HOMA-IR, homeostatic model assessment–insulin resistance; HR, hazard ratio; HU, Hounsfield unit; IMAT, intermuscular adipose tissue; PY, person-year; SAT, subcutaneous adipose tissue; SD, standard deviation; T2D, type 2 diabetes; TG, triglyceride; Waist C., waist circumference.

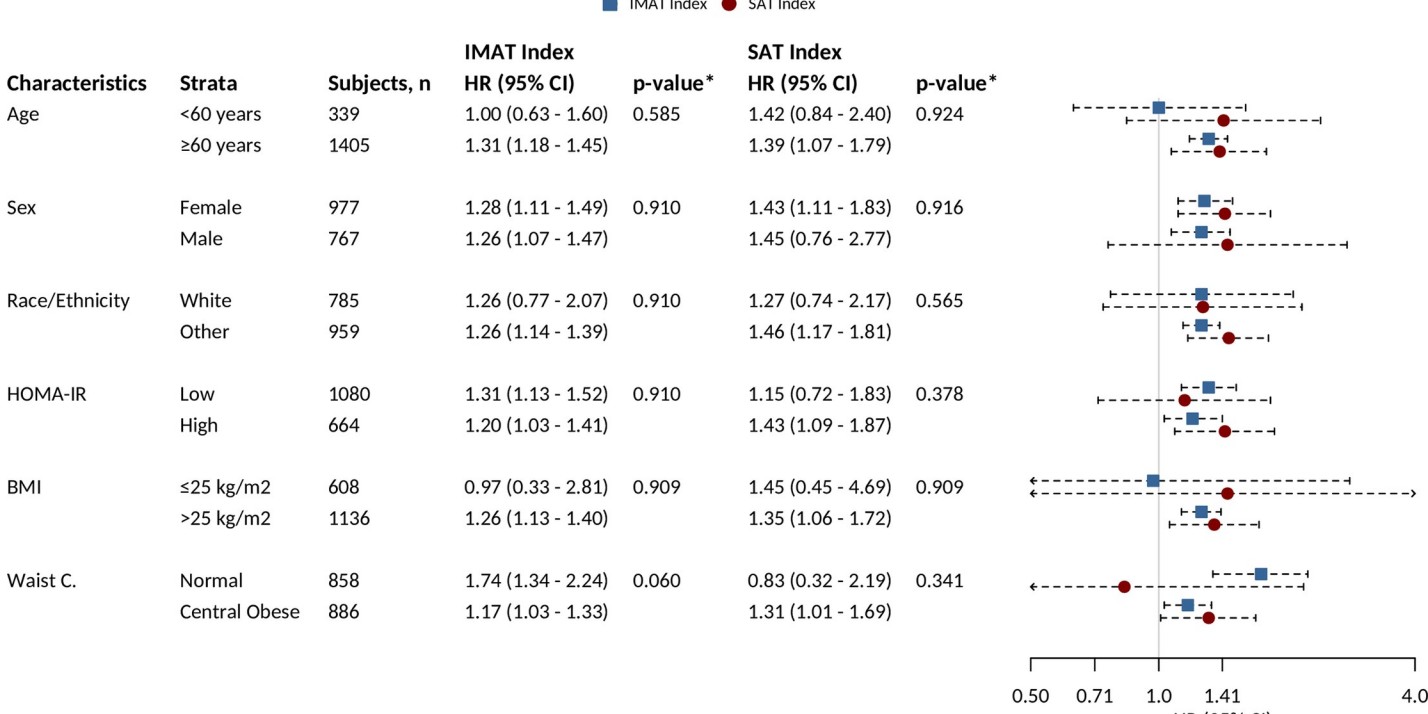

**Fig 2. Forest plot of the associations of adipose tissue biomarkers and T2D incidence by stratification variables.** *p-value for interaction. Models were adjusted for covariates in Model 1 (categorical age, sex, race/ethnicity, smoking status, alcohol drinking status, physical activity, TG, HDL cholesterol, and hypertension), except for the stratification variable. Reported p-values were corrected for multiple comparisons. BMI, body mass index; CI, confidence interval; HDL, high-density lipoprotein; HOMA-IR, homeostatic model assessment–insulin resistance; HR, hazard ratio; IMAT, intermuscular adipose tissue; SAT, subcutaneous adipose tissue; T2D, type 2 diabetes; TG, triglyceride; Waist C., waist circumference.

traditional risk factors of T2D, HOMA-IR, or BMI and waist circumference failed to reject the null hypothesis for lack of any associations between PM density quartiles or 1-SD increment in the PM density and T2D incidence (**S3 Table**).

In the stratified analyses, our models showed similar associations between PM density and T2D incidence in the strata of the traditional risk factors of T2D, HOMA-IR, BMI, and waist circumference (**S9 Fig**).

## Sensitivity and supplementary analyses

In the sensitivity analysis, self-reported physician-diagnosed T2D (the second criterion) was confirmed with the use of insulin or oral hypoglycemic agents or FPG $\geq$126 mg/dL in the follow-up exam. Similar to the results in the main analysis, the IMAT index was associated with T2D incidence, after adjusting for the effects of traditional T2D risk factors, HOMA-IR, and clinical anthropometric indices (i.e., BMI and waist circumference, **S4 Table**).

In a supplementary analysis in participants with prediabetes at baseline, the models failed to reject the null hypothesis for lack of associations between adipose tissue biomarkers and T2D incidence (**S5 Table**).

## Discussion

In this study, we investigated the associations between CT-derived adipose tissue biomarkers and T2D incidence over a median of 6.8 years in a population-based multiethnic cohort of

normoglycemic participants. We showed that high IMAT indices were associated with T2D incidence. The association was attenuated after accounting for effects of traditional T2D risk factors, HOMA-IR, and clinical anthropometric indices (i.e., BMI and waist circumference) but remained statistically significant.

The burden attributable to T2D can be potentially avoidable through primary and secondary preventive measures in the early stages of the disease [29]. Despite the robust literature on the predictive roles of genetic, cultural, behavioral, and environmental risk factors in T2D incidence [30,31], there is a paucity of research on the association between imaging biomarkers and T2D incidence. In routine clinical practice, chest CT exams are commonly used for various cardiopulmonary indications [32,33]. Specifically, chest CT exams are well-established tools for screening for coronary artery plaques and coronary calcium scoring in participants with intermediate 10-year atherosclerotic cardiovascular disease (ASCVD) risk [13] as well as for annual screening for lung cancer in high-risk adults [14]. All these CT exams retain data on adipose depots distribution, and there is the opportunity to extract CT-derived adipose depots biomarkers from these exams.

We showed that IMAT in chest CT exams (i.e., high IMAT index) is associated with T2D incidence. The IMAT index in chest CT exams may reflect the overall (upper and lower bodies) deposition of extramyocellular adipose depots, which is associated with insulin resistance [17,34,35], and, possibly, T2D incidence. The IMAT is located close to the muscle fibers and may play an intermediary role in insulin resistance through secreting pro-inflammatory cytokines, extracellular matrix proteins, and increasing local free fatty acids, or collectively, through altering the skeletal muscle microenvironment [34,35]. High BMI (or other clinical anthropometric indices) may be cited as a confounding factor in the association between this biomarker and the T2D incidence [36,37]. However, the findings of this study showed that when the effects of these clinical anthropometric indices and the IMAT are both accounted for in the models, the IMAT index remained associated with T2D incidence.

In this study, we have demonstrated that subcutaneous adipose depots in chest CT exams (i.e., high SAT index) were associated with T2D incidence. This finding was in line with prior studies on this topic, which suggested that (upper body) subcutaneous adipose depots on the chest CT exams may be associated with adverse cardiometabolic risk factors [38]. However, SAT index correlated with BMI and obesity, and the association between this CT-derived adipose tissue biomarker and T2D incidence was (at least partly) due to the role of high BMI (the association between SAT index and T2D incidence was attenuated toward the null after adjusting for effects of BMI and waist circumference). We also did not show any associations between intramyocellular lipid content (i.e., PM density) and T2D incidence.

Compared to preceding works [8–10], in this study, the adipose tissue biomarkers were obtained from the chest CT exams that were primarily performed to assess the lung parenchymal structure. The use of these exams shows the fact that the CT-derived adipose tissue biomarkers can be extracted opportunistically from the commonly performed chest CT exams for routine cardiopulmonary clinical indications, i.e., coronary calcium scoring and lung cancer screening. Although future studies are required to confirm the findings of this study, our study builds on the recommendation of clinical guidelines on the use of chest CT exams for coronary calcium scoring and lung cancer screening and can extend the value of these CT exams. The 2018 ACC/AHA Cholesterol Guideline recommends using chest CT exams for coronary artery calcium scoring in adults without T2D for making decisions about statin therapy [39]. The CT-derived adipose tissue biomarkers can be extracted from these CT exams and may be used in recommending T2D-related preventive measures in addition to potentially making decisions about statin therapy.

This study has a few but important limitations. This population-based study was observational, and, therefore, our findings are limited by the lack of interventions to control for potential effects of residual confounders. Moreover, this study was nested within the MESA, which is primarily designed to study cardiovascular diseases and outcomes. Contrary to the majority of cardiovascular diseases, T2D diagnosis is usually made in outpatient settings, and the review of the hospital records may not be able to capture all T2D diagnoses. To address this limitation, we supplemented the review of the hospital records with self-reported physician-diagnosed T2D. Although the potential inconsistencies in the timing of the self-reported physician-diagnosed T2D may have confounded our findings, we tested the possible effects of outliers on the observed results using a sensitivity analysis. Finally, the change of CT-derived adipose tissue biomarkers over the follow-up may be a potential source of unmeasured confounding effects in this study. The trajectory of these biomarkers may provide a better understanding of their association with T2D incidence.

In conclusion, this study showed an association between IMAT at baseline and T2D incidence over the follow-up in normoglycemic participants and suggested the potential role of intermuscular adipose depots in the pathophysiology of T2D.

## Supporting information

**S1 Checklist. STROBE Statement.** STROBE, Strengthening the Reporting of Observational Studies in Epidemiology.
(DOCX)

**S1 Table. Relevant Codes of ICD-9 and ICD-10.** ICD, International Classification of Diseases.
(DOCX)

**S2 Table. Statistical analyses.** SD, standard deviation.
(DOCX)

**S3 Table. Associations of PM density and T2D incidence.** Model 0: unadjusted. Model 1: adjusted for categorical age, sex, race/ethnicity, smoking status, alcohol drinking status, physical activity, TG, HDL cholesterol, and hypertension. Reported $p$-values were corrected for multiple comparisons. BMI, body mass index; CI, confidence interval; HDL, high-density lipoprotein; HOMA-IR, homeostatic model assessment–insulin resistance; HR, hazard ratio; HU, Hounsfield unit; PM, pectoralis muscle; PY, person-year; SD, standard deviation; T2D, type 2 diabetes; TG, triglyceride; Waist C., waist circumference.
(DOCX)

**S4 Table. Associations of adipose tissue biomarkers and T2D incidence (sensitivity analysis).** Model 0: unadjusted. Model 1: adjusted for categorical age, sex, race/ethnicity, smoking status, alcohol drinking status, physical activity, TG, HDL cholesterol, and hypertension. In this sensitivity analysis, self-reported physician-diagnosed T2D (the second criterion) was confirmed with the use of insulin or oral hypoglycemic agents or FPG ≥126 mg/dL in the follow-up exam. Participants with self-reported T2D but missing information on the use of insulin or oral hypoglycemic agents or FPG in the follow-up exam ($n$ = 17) were excluded. Reported $p$-values were corrected for multiple comparisons. BMI, body mass index; CI, confidence interval; FPG, fasting plasma glucose; HDL, high-density lipoprotein; HOMA-IR, homeostatic model assessment–insulin resistance; HR, hazard ratio; HU, Hounsfield unit; IMAT, intermuscular adipose tissue; PM, pectoralis muscle; PY, person-year; SAT, subcutaneous adipose tissue; SD, standard deviation; T2D, type 2 diabetes; TG, triglyceride; Waist C., waist circumference.
(DOCX)

**S5 Table. Associations of adipose tissue biomarkers and T2D incidence in participants with prediabetes (supplementary analysis).** *Models did not meet the proportional hazard assumption. Model 0: unadjusted. Model 1: adjusted for categorical age, sex, race/ethnicity, smoking status, alcohol drinking status, physical activity, TG, HDL cholesterol, and hypertension. In this supplementary analysis, Cox proportional hazard models were used to study the associations between adipose tissue biomarkers and T2D incidence in participants with prediabetes at baseline. Reported *p*-values were corrected for multiple comparisons. BMI, body mass index; CI, confidence interval; HDL, high-density lipoprotein; HOMA-IR, homeostatic model assessment–insulin resistance; HR, hazard ratio; HU, Hounsfield unit; IMAT, intermuscular adipose tissue; PM, pectoralis muscle; PY, person-year; SAT, subcutaneous adipose tissue; SD, standard deviation; T2D, type 2 diabetes; TG, triglyceride; Waist C., waist circumference.
(DOCX)

**S1 Fig. Flow diagram of the MESArthritis Ancillary Study.** CT, computed tomography.
(TIF)

**S2 Fig. Timeline of the MESArthritis Ancillary Study.** MESA, Multi-Ethnic Study of Atherosclerosis.
(TIF)

**S3 Fig. IMAT, SAT, and PM in the chest CT exam of a participant.** The chest CT exam of a 69-year-old normoglycemic male participant is shown here. **(A)** The coronal reconstruction of the CT exam is shown. The blue dashed line indicates the slice above the superior aspect of the aortic arch. **(B)** The area within the PM with attenuation below an individualized threshold (−89 HU in this participant) was measured as the IMAT (red). **(C)** The area between the PM and skin surface was measured as the SAT (red). **(D)** The density of PM (red) was measured as the mean HU of the pixels in the PM (after removing the pixels in the IMAT). CT, computed tomography; HU, Hounsfield unit; IMAT, intermuscular adipose tissue; PM, pectoralis muscle; SAT, subcutaneous adipose tissue.
(TIF)

**S4 Fig. Pattern of missing values.** In the dataset, there were missing data points in the HOMA-IR (161 [9.2%] data points), physical activity (13 [0.7%] data points), smoking status (10 [0.6%] data points), alcohol drinking status (8 [0.5%] data points), TG, HDL cholesterol, systolic and diastolic blood pressures, BMI, and waist circumference (2 [0.1%] data points in each). In this figure, the blue bars show number of participants with missing data points in the covariates marked with red circles below each bar. BMI, body mass index; HDL, high-density lipoprotein cholesterol; HOMA-IR, homeostatic model assessment–insulin resistance; TG, triglyceride; Waist C., waist circumference.
(TIF)

**S5 Fig. Symmetric correlation matrix for adipose tissue biomarkers and traditional T2D risk factors.** The blue color is used to display the positive correlations, and the red color is used for negative correlations. The attenuation of the color is proportional to the estimated Pearson correlation coefficients (numbers in the boxes). BMI, body mass index; FPG, fasting plasma glucose; HDL, high-density lipoprotein; HOMA-IR, homeostatic model assessment–insulin resistance; IMAT, intermuscular adipose tissue; PM, pectoralis muscle; SAT, subcutaneous adipose tissue; T2D, type 2 diabetes; TG, triglyceride; Waist C., waist circumference.
(TIF)

**S6 Fig. Associations of adipose tissue biomarkers and T2D incidence.** Smooth function estimates (red lines) obtained from fitting a generalized additive Cox proportional hazard models with integrated smoothness estimation on the dataset, with estimated 95% CI (blue lines) is shown. The results are reported on the scale of the adipose tissue biomarkers (per 1-SD increment), and the models were adjusted for covariates in Model 1 (categorical age, sex, race/ethnicity, smoking status, alcohol drinking status, physical activity, TG, HDL cholesterol, and hypertension). The numbers in brackets in the captions are the estimated degrees of freedom of the smooth curves. The rug marks along the x-axis indicate the adipose tissue biomarkers values. CI, confidence interval; HDL, high-density lipoprotein; HOMA-IR, homeostatic model assessment–insulin resistance; IMAT, intermuscular adipose tissue; PM, pectoralis muscle; SAT, subcutaneous adipose tissue; SD, standard deviation; T2D, type 2 diabetes; TG, triglyceride.
(TIF)

**S7 Fig. Forest plot of the association of IMAT index and T2D incidence by stratification variables.** *$p$-value for interaction. Models were adjusted for covariates in Model 1 (categorical age, sex, race/ethnicity, smoking status, alcohol drinking status, physical activity, TG, HDL cholesterol, and hypertension), except for the stratification variable. Reported $p$-values were corrected for multiple comparisons. BMI, body mass index; CI, confidence interval; HDL, high-density lipoprotein; HOMA-IR, homeostatic model assessment–insulin resistance; HR, hazard ratio; IMAT, intermuscular adipose tissue; T2D, type 2 diabetes; TG, triglyceride; Waist C., waist circumference.
(TIF)

**S8 Fig. Forest plot of the association of SAT index and T2D incidence by stratification variables.** *$p$-value for interaction. Models were adjusted for covariates in Model 1 (categorical age, sex, race/ethnicity, smoking status, alcohol drinking status, physical activity, TG, HDL cholesterol, and hypertension), except for the stratification variable. Reported $p$-values were corrected for multiple comparisons. BMI, body mass index; CI, confidence interval; HDL, high-density lipoprotein; HOMA-IR, homeostatic model assessment–insulin resistance; HR, hazard ratio; SAT, subcutaneous adipose tissue; T2D, type 2 diabetes; TG, triglyceride; Waist C., waist circumference.
(TIF)

**S9 Fig. Forest plot of the association of PM density and T2D incidence by stratification variables.** *$p$-value for interaction. Models were adjusted for covariates in Model 1 (categorical age, sex, race/ethnicity, smoking status, alcohol drinking status, physical activity, TG, HDL cholesterol, and hypertension), except for the stratification variable. Reported $p$-values were corrected for multiple comparisons. BMI, body mass index; CI, confidence interval; HDL, high-density lipoprotein; HOMA-IR, homeostatic model assessment–insulin resistance; HR, hazard ratio; PM, pectoralis muscle; SD, standard deviation; T2D, type 2 diabetes; TG, triglyceride, Waist C., waist circumference.
(TIF)

## Acknowledgments

The authors thank the other investigators, the staff, and the participants of the MESA study for their valuable contributions. A full list of participating MESA investigators and institutions can be found at MESA-NHLBI.org.

## Author Contributions

**Conceptualization:** Farhad Pishgar, Mahsima Shabani, Thiago Quinaglia A. C. Silva, David A. Bluemke, Matthew Budoff, R Graham Barr, Matthew A. Allison, Alain G. Bertoni, Wendy S. Post, João A. C. Lima, Shadpour Demehri.

**Data curation:** Farhad Pishgar, Shadpour Demehri.

**Formal analysis:** Farhad Pishgar.

**Investigation:** Farhad Pishgar, Mahsima Shabani, Thiago Quinaglia A. C. Silva, David A. Bluemke, Matthew Budoff, R Graham Barr, Matthew A. Allison, Alain G. Bertoni, Wendy S. Post, João A. C. Lima, Shadpour Demehri.

**Methodology:** Farhad Pishgar, Mahsima Shabani, Thiago Quinaglia A. C. Silva, David A. Bluemke, Matthew Budoff, R Graham Barr, Matthew A. Allison, Alain G. Bertoni, Wendy S. Post, João A. C. Lima, Shadpour Demehri.

**Project administration:** João A. C. Lima, Shadpour Demehri.

**Resources:** João A. C. Lima, Shadpour Demehri.

**Software:** Farhad Pishgar, Shadpour Demehri.

**Supervision:** David A. Bluemke, Matthew Budoff, R Graham Barr, Matthew A. Allison, Alain G. Bertoni, Wendy S. Post, João A. C. Lima, Shadpour Demehri.

**Validation:** Farhad Pishgar, Shadpour Demehri.

**Visualization:** Farhad Pishgar.

**Writing – original draft:** Farhad Pishgar, Shadpour Demehri.

**Writing – review & editing:** Farhad Pishgar, Mahsima Shabani, Thiago Quinaglia A. C. Silva, David A. Bluemke, Matthew Budoff, R Graham Barr, Matthew A. Allison, Alain G. Bertoni, Wendy S. Post, João A. C. Lima, Shadpour Demehri.

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
