## [Editor Report · Decision Letter 0]

3 Feb 2021

Dear Dr Demehri, 

Thank you for submitting your manuscript entitled "Adipose tissue biomarkers and risk of type 2 diabetes incidence in normoglycemic participants:  The MESArthritis Ancillary Study" for consideration by PLOS Medicine.

Your manuscript has now been evaluated by the PLOS Medicine editorial staff as well as by an academic editor with relevant expertise and I am writing to let you know that we would like to send your submission out for external peer review.

Please re-submit your manuscript within two working days, i.e. by February 8, 2021.

Kind regards,

Beryne Odeny

Associate Editor

PLOS Medicine

---

## [Decision Letter · Decision Letter 1]

18 Mar 2021

Dear Dr. Demehri,

Thank you very much for submitting your manuscript "Adipose tissue biomarkers and risk of type 2 diabetes incidence in normoglycemic participants:  The MESArthritis Ancillary Study" (PMEDICINE-D-21-00345R1) for consideration at PLOS Medicine. 

[LINK]

In light of these reviews, I am afraid that we will not be able to accept the manuscript for publication in the journal in its current form, but we would like to consider a revised version that addresses the reviewers' and editors' comments. Obviously we cannot make any decision about publication until we have seen the revised manuscript and your response, and we plan to seek re-review by one or more of the reviewers. 

We expect to receive your revised manuscript by Apr 08 2021 11:59PM. Please email us (plosmedicine@plos.org) if you have any questions or concerns.

We look forward to receiving your revised manuscript. 

Sincerely,

Beryne Odeny, 

PLOS Medicine 

plosmedicine.org

- Please revise your title according to PLOS Medicine's style. Your title must be nondeclarative and not a question. It should begin with main concept if possible. "Effect of" or “risk of” should be used only if causality can be inferred, i.e., for an RCT. Please place the study design ("A retrospective study,") in the subtitle (i.e., after a colon). For example, “Adipose tissue biomarkers and type 2 diabetes incidence in normoglycemic participants in The MESArthritis Ancillary Study: A retrospective cohort study”

- In the Data Availability Statement, please provide a direct email address or more direct weblink which can be used to access the data. Clicking on the provided link, it is not clear which steps need to be taken to request data access.

- Abstract:

1. Please ensure that all numbers presented in the abstract are present and identical to numbers presented in the main manuscript text.

2. Please include the study design, population and setting, number of participants, years during which the study took place, length of follow up, and main outcome measures.

3. Please include the actual amounts and/or absolute risk(s) of relevant outcomes, not just relative risks or correlation coefficients 

4. Please include a summary of adverse events if these were assessed in the study.

5. In the last sentence of the Abstract Methods and Findings section, please describe the main limitation(s) of the study's methodology.

6. Abstract Conclusions: Please address the study implications without overreaching what can be concluded from the data; the phrase "In this study, we observed ..." may be useful. Please interpret the study based on the results presented in the abstract, emphasizing what is new.

- Please conclude the Introduction with a clear description of the study question or hypothesis.

- For this observational study, in the manuscript text, please indicate: (1) the analytical methods by which you planned to test your hypothesis, (2) the analyses you actually performed, and (3) when reported analyses differ from those that were planned, transparent explanations for differences that affect the reliability of the study's results. If a reported analysis was performed based on an interesting but unanticipated pattern in the data, please be clear that the analysis was data-driven.

- Did your study have a prospective protocol or analysis plan? Please state this (either way) early in the Methods section. 

- Please ensure that the study is reported according to the STROBE guideline, and include the completed STROBE checklist as Supporting Information. Please add the following statement, or similar, to the Methods: "This study is reported as per the Strengthening the Reporting of Observational Studies in Epidemiology (STROBE) guideline (S1 Checklist)." The STROBE guideline can be found here: http://www.equator-network.org/reporting-guidelines/strobe/

- Your study is observational and therefore causality cannot be inferred. Please remove language that implies causality, such as - predict, risk, effect, etc. Instead, refer to associations consistently throughout the text.

- Please define the length of follow up (eg, in mean, SD, and range).

- How was race/ethnicity defined and by whom? Why was race/ethnicity considered important in this study and what it is believed to represent e.g., are SES or genetic differences being attributed to race/ethnicity?

- Please indicate and discuss whether your findings were validated in a different data set.

- In statistical methods, please refer to any post-hoc corrections to correct for multiple comparisons during your statistical analyses. If these were not performed please justify the reasons. Please refer to our statistical reporting guidelines for assistance (https://journals.plos.org/plosone/s/submission-guidelines.#loc-statistical-reporting)

- We note that your data are clustered at hospital and regional levels. Please elaborate how you account for clustering in your statistical methods.

- Please describe how you selected your adjustment variables. 

- We note the potential for unobservable confounding in this observational study. Please consider using robust methods such as propensity score matching to address this.

- Please write “95% CI” when you provide estimates and 95% CI

- Please specify the significance level used (eg, P<0.05, two-sided) and the statistical test used to derive a p value.

- Please provide the full name(s) of the institutional review board(s) that provided ethical approval.

- For S3 figure of the chest CT, please ensure that you have complied with our figures requirements http://journals.plos.org/plosmedicine/s/figures.

- Please replace “nondiabetics” with “persons without diabetes.”

- Please replace "Caucasian" with "white" throughout the paper.

Comments from the reviewers:

Reviewer #1: The authors examined the associations of CT-measured indices with incident type 2 diabetes (T2D). This study has a novel component of the use of CT-derived variables. Its contributions to science and clinical practice may be meaningful, while the reviewer is not an expert on it. While positive remarks are possible, some concerns are present as described in the following paragraphs: Major Comments and Minor Comments.

Major comments:

0.

The authors identified a significant interaction by BMI for the primary finding. Thus, the average association presented in the abstract and the main results is not meaningful. The abstract and "Author Summary" should clarify that the positive association was present among those with overweight or obesity (BMI>24.9).

The authors need to assess the association stratified by BMI with statistical adjustment for BMI (within stratum) and waist circumference. It is not clear if the authors did it.

1. The authors should revise the description of the results throughout the manuscript. The authors dichotomized their inference into two mainly: significant or not. That is malpractice in recent medical science. The authors should avoid it throughout the manuscript.

Accounting for no perfect measure of central or whole-body adiposity, the authors need to keep the following interpretations:

i) there were associations of the SAT and IMAT indices with T2D among the participants with overweight or obesity.

ii) the association was partly related to insulin resistance and anthropometric measures.

iii) The intermuscular fat related to the development of T2D. Thus, this study demonstrated that the possible failure of storing fat intracellularly in adults with overweight or obesity could be the consequence, pathological process, or both of the development of T2D.

The authors should avoid describing that one type of association was independent of the adiposity or that the IMAT index predicted T2D. This kind of interpretation would be too strong given the observational analyses with measurement errors. Of note, even if the association became null after adjustment for BMI and waist circumference, this study seems to be pathophysiologically important. However, for the particular argument, interaction by overweight or obesity status is essential to highlight.

2. The introduction and discussion are insufficient.

The authors should cite several genetic, Mendelian randomization studies relating anthropometric measures to T2D, such as the one in JAMA (LA Lotta et al., JAMA, 2018). Genetic evidence is already available to indicate that fat storage and its location matter in the development of T2D. The authors should present it as background information for this work and discuss the pathophysiology of T2D. 

3. The reviewer has considered the pathophysiological importance of this work. The authors emphasized it in the introduction and because the reviewer has recognized the sparsity of the relevant evidence in large-scale epidemiological research. However, the authors may eagerly want to establish the importance of the CT-derived measures to predict T2D risk regardless of the etiological pathway. If so, the authors should revise the introduction to explain why the prediction modelling for T2D should be improved, citing previous studies on T2D prediction (e.g. ones from Framingham Offspring Study, ARIC, MESA). The authors have not done it, leaving the argument about prediction unconvincing or not structured well. 

Moreover, if the authors are interested in prediction, its evaluation requires refinement. The authors mainly interpreted whether the measures of associations were significant or not. This interpretation was substandard for its utility for the prediction. A further quantitative evaluation scoping possible clinical actions will be essential (see Cook et al., Annals Intern Med, 2009;150(11):795-802).

Finally, if the authors want to argue the utility of a CT exam for the prediction of T2D, the evaluation should include other CT results in addition to the chest ones. The emphasis for prediction would not make sense in this manuscript.

4. 

S7 Fig is incorrect. Hazard ratios cannot be negative. If an author estimates incident rate ratio or hazard ratio, one reference point without any standard errors should be available. The current figure does not have it. A valid explanation is necessary if the figure is valid. 

Minor comments:

The primary results are owing to an assessment of statistical significance applied to p-value=0.036. Given the number of tests operated in this study (for the two CT measures, for instance), the inference based on the p-value is inappropriate. As generally recommended against the dichotomic use of p-value, the authors should revise the manuscript to judge the results based on statistical significance.

Abstract: 

Background: "and contribute to" should be deleted. After the deletion, the sentence still works, and it will turn out to be sharp and straightforward.

Calendar years of the follow-up start and finish should be available in the abstract. The MESA started around 2000, but this particular study's baseline started around 2010 from the CT measurements. Thus, for some readers, years should be documented.

Use "T2D" after spell out once the authors explain the abbreviation throughout the abstract.

The authors should not provide a positive or negative connotation based on statistical significance. The authors have stated, "The association of the IMAT index remained statistically significant after adjusting for body mass index and waist circumference (HR:1.30 [95%CI: 1.02-1.65]), however, the SAT index was no longer associated with type 2 diabetes incidence." This sentence should be revised as follows: "The associations of the SAT and IMAT indices with T2D incidence were attenuated toward the null after adjustment for body mass index and waist circumference, with HR (95% CI) of X.XX (X.XX-X.XX) and 1.30 (1.02-1.65), respectively".

Given the possible measurement errors of body mass and waist circumference, the authors should consider a more substantial attenuation after the adjustment and describe the results and interpretation, accounting for the consideration.

The conclusion is too strong and odd. The authors must not be interested in the improvement of prediction via the index they used. Instead, the authors must be interested in how important the intermuscular fat may be in the pathophysiology of T2D. In a sense, the authors may not want to digress the logical flow to a prediction story. The authors may want to state the association partly related to adiposity and highlight the etiological importance of the intermuscular fat. The conclusion requires such a revision. 

Author Summary:

The first text read as if the participants had used time-to-event data. The authors can delete "using longitudinal time-to-event analysis". By saying "baseline" and "incidence", the longitudinal aspect is readable.

Delete "in this study" and start the bullet point with "We".

Line 75: The authors should avoid the dichotomic expression of the results.

　

Methods

Line 186-187: For additive modelling and smoothness estimation, the statistical function should be clear enough for readers to replicate the analysis. There can be different ways.

Line 188: "consecutive nested" is not necessary unless this phrase means something extraordinary.

BMI and alcohol can have non-linear associations with T2D incidence. The current approach would be suboptimal. The authors are, in particular, interested in the role of BMI in the evaluation of the association. Therefore, additive modelling for BMI is reasonable in this study.

Results:

The authors should document the increase of standard errors or confidence intervals after adjusting BMI and waist circumference. Table 2 clarifies the collinearity of the SAT with BMI and waist circumference.

Delete S6 Fig or delete S2 Table and include the numeric info into S6 Fig.

Line 228-229. There was no apparent cubic association. Also, the authors did not formally test whether or not there was a cubic or non-linear association. The inflexion point the authors described did not reflect an objective assessment. The authors should delete the sentence without any statistical approach to identify such a value. Also, log-2 values are hard to interpret. The authors should convert it to a raw value to facilitate an interpretation when needed.

The authors should describe the results accounting for the interaction by BMI. It is critically important to state the results after the stratification and after adjusting BMI and waist circumference.

Discussion.

The first paragraph is misleading. The association was present among those with overweight and obesity. This should be the basis to characterize the association.

The authors' discussion is appropriate for the potential use of CT scanning for the prediction of T2D and other diseases. However, this study is too far from the argument because the authors did not interpret the results quantitatively for the clinical application. Also, this study identified the interaction by BMI and therefore has not granted the wide application of the CT scanning to a general population.

The reviewer is reserving other minor comments.

Reviewer #2: Statistical review

This paper reports an observational study that assessed association between biomarkers and risk of developing diabetes.

I had some comments on the statistical methods and reporting, which are provided below.

1. Abstract: I'd recommend providing the estimated HR and 95%CI for adjusted SAT association.

2. Abstract line 62 "can predict a" - I recommend slightly less causal language such as 'is associated with'.

3. Abstract/Methods/Results: Were IMAT and SAT included in the same adjusted model? It would be useful to note whether IMAT was an independent risk variable after adjustment for SAT.

4. Line 197 - I'd recommend adding how many patients were excluded from the complete cases analysis. I think Figure S5 requires more information in the caption as it wasn't clear to me how to interpret it currently.

5. Statistical analysis: it would be useful to provide a pre-specified analysis plan or otherwise note which analyses described here were specified prior to the data being available.

6. Line 188-190: I did not follow what the authors meant by 'consecutive nested … were used to estimate' - is this some type of model building procedure? More details on this would be useful.

7. Line 199 - did the stratified analyses also adjust for the variables in the previous paragraph?

8. Line 222 - I'd recommend clarifying the 74 does not include the 17 self-identified participants mentioned on line 160?

9. Line 266 - I would add results from the Intramuscular adiposity section to the abstract as currently it looks a bit selective to report only two of the three main sets of results. I'd also add the results on this to table 2 from the supplementary table.

James Wason

Reviewer #3: See file

[LINK]

---

## [Decision Letter · Decision Letter 2]

10 Jun 2021

Dear Dr. Demehri,

Thank you very much for re-submitting your manuscript "Adipose tissue biomarkers and type 2 diabetes incidence in normoglycemic participants in The MESArthritis Ancillary Study: A retrospective cohort study" (PMEDICINE-D-21-00345R2) for review by PLOS Medicine.

I have discussed the paper with my colleagues and the academic editor and it was also seen again by three reviewers. I am pleased to say that provided the remaining editorial and production issues are dealt with we are planning to accept the paper for publication in the journal.

[LINK]

We look forward to receiving the revised manuscript by Jun 17 2021 11:59PM.   

Sincerely,

Beryne Odeny, 

Associate Editor 

PLOS Medicine

plosmedicine.org

Requests from Editors:

1. Please remove the word “retrospective” from the title

2. Please use the "Vancouver" style for reference formatting and see our website for other reference guidelines. For example:

a. Please ensure that journal name abbreviations match those found in the National Center for Biotechnology Information (NCBI) databases, and are appropriately formatted and capitalized. https://journals.plos.org/plosmedicine/s/submission-guidelines#loc-references. For example, in references # 37, 38 etc., the journal titles need to be abbreviated in line with the Vancouver style.

b. Please remove excess text from Reference # 6, “AstraZeneca, MedImmune, Novo Nordisk, and ERX Pharmaceuticals and grants from Aegerion outside the submitted work. Dr Scott is an employee and shareholder in GlaxoSmithKline. Dr Burgess reported grants from Wellcome Trust/Royal Society and the UK Medical Research Council during the conduct of the study. No other disclosures were reported”

Comments from Reviewers:

Reviewer #1: The authors have revised the manuscript well. Some minor comments are come up with as the followings.

In the imputation, the authors stated the use of "Multivariate Imputation by Chained Equations (MICE)". The name is correct, according to the publication referred to by the authors. However, the approach has now been well-known as "multiple imputation by chained equations (MICE)". Technically, both are correct, but the latter is standard as far as the Reviewer reviewed some publications. To avoid a potentially negative impression readers may come up with, the authors may want to avoid stating the procedure's name: for example, to state, "imputing missing values with chained equations and predictive mean matching method".

In a couple of tables, p-values presented appeared to be after adjustment for false discovery rate. Some readers would be confused, and therefore the application of the FDR should be noted in the footnote wherever appropriate. 

S4 Table was not clear what was different from the main table. The authors should confirm that each table and figure stands alone so that readers can understand the material. Also, person-time values across categories should be available. Otherwise, future reviewers cannot use the statistics for meta-analysis readily.

In the discussion section, the authors should discuss the possible mechanism of the authors' findings. The current statement is insufficient and very naive (Page 18). The authors should discuss why the IMAT of the chest showed an association with T2D incidence. For example, mechanisms of T2D development include skeletal muscle insulin resistance, hepatic insulin resistance, and beta-cell dysfunction. Then, it is not explicitly clear why the chest IMAT relates to the incidence of T2D. General correlates with IMAT around any tissues are unhelpful.

Reviewer #2: Thank you to the authors for addressing my previous comments well. I have no further issues to raise.

Reviewer #3: I Think the authors have addressed all questions adequately and I have no further comments.

[LINK]

---

## [Editor Report · Decision Letter 3]

16 Jun 2021

Dear Dr Demehri, 

On behalf of my colleagues and the Academic Editor, Dr. Weiping Jia, I am pleased to inform you that we have agreed to publish your manuscript "Adipose tissue biomarkers and type 2 diabetes incidence in normoglycemic participants in The MESArthritis Ancillary Study: A cohort study" (PMEDICINE-D-21-00345R3) in PLOS Medicine.

PRESS

Sincerely, 

Beryne Odeny 

Associate Editor 

PLOS Medicine